# The Frequency of a Magnetic Field Reduces the Viability and Proliferation of Numerous Tumor Cell Lines

**DOI:** 10.3390/biom15040503

**Published:** 2025-03-31

**Authors:** Isabel López de Mingo, Marco Xavier Rivera González, Milagros Ramos Gómez, Ceferino Maestú Unturbe

**Affiliations:** 1Escuela Técnica Superior de Ingenieros de Telecomunicación (ETSIT), Universidad Politécnica de Madrid, 28223 Madrid, Spain; milagros.ramos@ctb.upm.es; 2Centro de Tecnología Biomédica (CTB), Universidad Politécnica de Madrid (UPM), Avda. Campus de Montegancedo, M40 Km38, Pozuelo de Alarcón, 28223 Madrid, Spain; marco.rivera@ctb.upm.es (M.X.R.G.); ceferino.maestu@ctb.upm.es (C.M.U.); 3Escuela Técnica Superior de Ingenieros Informáticos (ETSIINF), Universidad Politécnica de Madrid, 28223 Madrid, Spain; 4Centro de Investigación Biomédica en Red (CIBER-BBN), 28029 Madrid, Spain

**Keywords:** tumor cells, ELF-EMFs, magnetic field, frequency, viability, proliferation

## Abstract

The frequency of a magnetic field has led to the development of physicochemical interaction models and to the consideration of the role of frequency as a fundamental parameter in the change of cell behavior. The main objective of this article is to find a frequency window to decrease the viability and proliferation of different tumor cell lines to compare the frequency response of each. For this purpose, tumor cell lines PC12 (rat), B16F10 (mouse), SKBR3 (human), MDA-MB-231 (human), and the non-tumor cell line 3T3 (mouse) are exposed to a magnetic field of 100 µT for 24, 48, and 72 hours in frequency windows contained in the range [20–100] Hz, and their viability and proliferation behavior is evaluated. The results show a frequency-, exposure-time-, and cell-line-dependent behavior, with the most pronounced changes for most cell lines at frequencies of 45, 50, and 55 Hz. It is concluded that each cell type could respond to specific frequency codes that allow the modification of its behavior in vital cellular processes related to tumor development. Knowledge of these codes would allow for the therapeutic application of magnetic fields in oncological pathologies.

## 1. Introduction

Cancer is a heterogeneous group of diseases at the molecular level in which cells divide continuously and excessively [1]. Tumor cells have been one of the most widely used biological models in bioelectromagnetism for two reasons. Firstly, there are numerous studies related to the evaluation of exposure to extremely low frequency electromagnetic fields (ELF-EMFs) at environmental or occupational levels with acute or chronic exposures to determine whether cellular processes that induce tumor formation or growth are affected, especially after the International Agency for Research in Cancer established ELF-EMFs in category 2B as a possible carcinogenic agent [2]. Second, many research groups focus on modulating the cellular response with the aim of reducing the proliferative capacity or viability of tumor cells with potential therapeutic applications.

Numerous studies have pointed to EMFs as possible inducers of tumorigenic processes at the cellular [3,4,5,6,7], animal [4,5,7,8], and individual level [4,9,10,11,12,13] for not specifying sufficient adaptation mechanisms to the introduction of a considerable amount of artificial electromagnetic fields [14]. The group of cells studied is always very heterogeneous with breast cancer [15,16,17,18,19,20,21,22,23,24,25,26], osteosarcoma [27,28,29,30], leukemia or lymphoma [18,20,31,32,33,34,35,36,37,38], melanoma [15,39], gastric tumor or colon cancer [15,37,40,41,42,43], pancreatic cancer [18,43,44], prostate carcinoma [20,45], renal adenocarcinoma [46,47], cervical cancer [45,48,49], or tumors related to the nervous system, such as gliomas and neuroblastomas [18,26,30,44,50,51,52,53]. Most of these studies focus their results on cellular processes involved in the appearance and increase in the number of tumor cells, such as viability (defined as the ratio of live to dead cells in a population) [17,20,21,28,33,47], proliferation (defined as the measure of the number of cell divisions in a population) [32,36,37,48], or apoptosis (defined as programmed cell death by which damaged cells in a population are eliminated) [16,17,20,23,33,37,47,49]. When the regulatory mechanisms of these processes are altered, the tumor cell divides uncontrollably.

Since the 1980s, there have been numerous studies on the exposure of cellular models to magnetic fields, although the reported effects have been varied and sometimes contradictory, it has not been possible to establish a mechanism of interaction that can explain them [54]. Frequency, as an exposure parameter, has been one of the main actuators in many of the physical models related to resonance, such as Liboff’s cyclotron frequency [55,56,57], or the ion parametric resonance model proposed by Lednev [58]. Ross Adey first introduced the term “window” after three independent laboratories simultaneously found cellular effects responding to a resonance mechanism at certain values of frequency and intensity at which the biological response of the system was more pronounced than in the rest of the frequency and amplitude range [59]. This was of great importance for the consideration of frequency as the responsibility for biological responses observed in the experiment. Frequency, intensity, and time windows can be grouped under the term “biological window”, and are defined as a means by which electromagnetic fields interact with biological systems [60,61,62,63]. Despite the existence of results for in vitro models that suggest that frequency plays a major role in the alteration of cellular processes, the exposure regulations governing the limits to which the general population can be exposed are based on the “dose effect” or “more is worse” criterion in which intensity is the main driver and frequency takes on a mere secondary role [64].

The main objective of this research article is to find a specific frequency of a magnetic field applied at continuous intensity that allows for the reduction of viability and proliferation of different tumor cell models at different exposure times, for this purpose, these cellular processes are studied by performing a search based on the criterion of “bioactive window”. Furthermore, the results obtained are compared with those previously found in the cells of nervous tissue [65] to determine whether the response is dependent on the cell type used.

## 2. Materials and Methods

### 2.1. Cell Cultures

Cell lines from different species (human, rat, and mouse) and tissues (mammary, kidney, skin, and connective) are used for comparative purposes in response to the same magnetic field. Human HER2+ breast cancer cell lines SKBR3 (ATCC number: HTB-30) and triple negative MDA-MB-231 (ATCC number: CRM-HTB-26) were obtained from ATCC (American Type Culture Collection, LGC Standards, Teddington, UK). Adherent rat pheochromocytoma (PC12 Adh), mouse fibroblast (3T3), and murine melanoma (B16F10) cell lines were provided by the Instituto Cajal de Madrid belonging to the Consejo Superior de Investigaciones Científicas (CSIC). The MDA-MB-231, B16F10, and 3T3 lines were grown in a monolayer culture in Dulbecco’s modified Eagle medium with elevated glucose (DMEM) (DDBiolab, w/L-Glutamine, without sodium pyruvate, cat. no. L0102-500), supplemented with 10% fetal bovine serum (DDBiolab, cat. no. P30-3302, Barcelona, Spain), 1% L-Glutamine (DDBiolab, 200 mM, cat. no. P04-80100, Barcelona, Spain), and 1% penicillin/streptomycin (DDBiolab, penicillin 5000 Ul/mL, streptomycin 5, Barcelona, Spain). The SKBR3 cell line was grown in monolayer culture in Roswell Park Memorial Institute (RPMI) culture medium (DDBiolab, w/L-Glutamine, cat. no. L0500-500, Barcelona, Spain), supplemented with 10% fetal bovine serum (DDBiolab, cat. no. P30-3302) and 1% penicillin/streptomycin (DDBiolab, penicillin 5000 Ul/mL, streptomycin 5, Barcelona, Spain). The PC12 cell line was maintained in a monolayer culture in Dulbecco’s modified Eagle medium with elevated glucose (DMEM) (DDBiolab, w/L-Glutamine, no sodium pyruvate, cat. no. L0102-500, Barcelona, Spain), supplemented with 5% fetal bovine serum (DDBiolab, cat. no. P30-3302, Barcelona, Spain), 10% donor horse serum (DDBiolab, cat. no. S0900-500), 2mM L-glutamine (DDBiolab, 200 mM, cat. no. P04-80100, Barcelona, Spain), 25µG/mL gentamicin (DDBiolab, cat. no. L0011-010, Barcelona, Spain), and 2.5 µg/mL amphotericin B (DDBiolab, cat. no. P06-01050, Barcelona, Spain). All cell lines were cultured at 37 °C temperature under an atmosphere of 5% CO_2_ in air in Thermo Scientific 3111 series II incubators (Thermo Fisher Scientific Inc., Waltham, MA, USA). All cells used in the experiment had a pass lower than 15. Cell subpopulations were prepared using the cell passaging technique when the culture plates were close to 90% confluence.

### 2.2. Electromagnetic Field Exposure System

The exposure system used was specially designed at the Biolelectromagnetism Laboratory of the Centro de Tecnología Biomédica, Universidad Politécnica de Madrid, for use inside cell culture incubators, with high homogeneity of the magnetic field applied on the treated plate, as published previously (Figure 1) [66]. Briefly, two identical coils are used, called RILZ coils, with a capsule shape consisting of two semicircles with a radius of 5 cm joined by two 10-cm-long straight lines. The separation between the coils is 3.5 cm with a width of 7 cm in which the copper wires of AWG 18 are placed. Each of the coils has 222 turns of enameled copper. The coils are placed on plastic supports raised with respect to the metal tray of the incubator to avoid the noise generated by the induction of the magnetic field by direct contact with the grounded metal surface. The culture plates treated with the magnetic field are placed in the center of plastic supports that ensure that they are at the point of homogeneity of the system. The coil power electronics is also a proprietary design of the complete coil system, consisting of a microprocessor that generates a square signal. An LCD display shows the frequency and current values, and these are controlled by precision potentiometers connected to an analog-to-digital converter (ADC) of the microcontroller. A current between 0 and 2 Amps is set through a power MOSFET feeding the RILZ coils, and a current source controlled by the microcontroller. The frequency, between DC and 200 Hz, is set using interrupts generated by the microcontroller’s internal timer. Magnetic field measurements are made with the Lake Shore Model 480 fluxmeter (Lake Shore Cryotronics, Westerville, OH, USA) and a Model MMZ-2502-UH triaxial probe (Lake Shore, Cryotronics, OH, USA).). The absolute value of the pre-existing magnetic field in the three space directions is from x (23.14 ± 0.39 µT), y (42.30 ± 1.19 µT), and z (5.96 ± 0.49 µT).

### 2.3. Exposure Conditions

The exposure conditions for each of the tests are shown in Table 1 and Figure 2. For comparison purposes, the same exposure parameters as in a previous publication evaluating the effect of the magnetic field on nerve tissue cell lines are used [65]. A fixed intensity of 100 µT (exposure limit established by the recommendation 1999/519/EC in Spain for a frequency of 50 Hz) is established [67]. The frequency is variable according to the range explored, with a minimum value of 20 Hz and a maximum of 100 Hz. The exposure time is also variable, and experiments are carried out on viability for 24, 48, and 72 h, and the effects on proliferation are evaluated at 24 h. The applied waveform is square, as described in the previous publications [65,66].

Before seeding the experiments, the homogeneity area provided by the system was determined according to the plate used; a homogeneity value greater than 97% was maintained in all exposed samples [66]. For all experiments, seeding was performed 24 h before exposure.

The controls are plates kept in the same conditions as the treated plates but with the equipment turned off, so they are subject to the pre-existing magnetic field values already described.

### 2.4. Metabolic Activity Assay

Cells are seeded in a 96-well plate at a final concentration of 60,000 cells/mL. The 3-(4,5-dimethylthiazol-2-yl)-2,5-diphenyl-2H-tetrazolium bromide (MTT) assay (Biotium, MTT Cell Viability Assay Kit, cat. no. 30006, Fremont, CA, USA) is used, following the manufacturer’s instructions. Briefly, 10 µL of the MTT agent was added to each of the tested wells and the plate was incubated in the dark for 4 h in the incubator at 37 °C. After the time had elapsed, 200 µL of dimethyl sulfoxide (DMSO, Corning Media Tech, Cat. no. 15303671, New York, NY, USA) was added to each of the wells. They were resuspended to dissolve the formazan salts.

The absorbance was measured using a HEALES model MB-580 microplate reader (HEALES, Shenzhen, China) at wavelengths of 570 nm (MTT signal) and 630 nm (background signal). Each of the frequencies has a total of 10 replicates in 2 independent experiments, including controls. The result of the subtraction of absorbances of each of the replicates was ordered from lowest to highest for each of the frequencies, on the one hand, and, on the other, for their respective control. The viability percentages are calculated as follows according to the established increasing order:$${Viability\;\left(\%\right)}_{i}=\frac{{Abs}_{570–630\;nm}\;({Exp}_{i})}{{Abs}_{570–630\;nm}\;({Control}_{i})}$$

The percentage of viability will be represented as deviations of viability from the control. They are calculated as follows:$${Viability\;(\%)}_{j}={Viability\;(\%)}_{i}-100\%$$

Once the viability percentages were calculated, they were again ordered from lowest to highest, and the highest and lowest values of each set were discarded in order to homogenize the final percentages. These eight viability percentages are included in the statistical software. The graphical representation of the results obtained from viability is performed using the Prism 9 software (GraphPad Software, v.9.3.1., Boston, MA, USA).

The experimental design can be observed in Figure 3.

### 2.5. Proliferation

Proliferation assays are performed using Trypan Blue staining agent (DDBiolab, cat. no. P08-34100) and LUNA II automatic cell counter (Logos Biosystems, Anyang-si, Republic of Korea). Cells were seeded in 60 mm diameter cell culture plates with a final concentration of 300,000 cells/mL. Cells were harvested by the trypsinization technique. Briefly, after the exposure time had elapsed, the culture medium was removed from the plate, washed with 0.5 mL of 1X phosphate buffered saline PBS (DDBiolab, cat.no. SH30258.01), and 0.5 mL of 1X phosphate buffered saline PBS (DDBiolab, cat.no. SH30258.01) was added. After removal, 0.5 mL of trypsin enzyme (Thermo Fisher Scientific Inc., cat.no.15090046) was added and the plate was incubated at 37 °C for 5 min. After the time had elapsed, 1 mL of the culture medium was added to the plate and the contents were poured over a 15 mL Falcon tube and centrifuged for 5 min at 1500 rpm and 21 °C. The supernatant was removed and the cells were resuspended in 1 mL of the culture medium. An amount of 10 µL of tube contents after resuspension and 10 µL of staining agent are removed and mixed. Of the 20 µL of the mixture, 12 µL are used to load the counting chamber, which is inserted into the automatic counter. The counter returns information on the number of live and dead cells. The experiments are performed in triplicate in independent experiments. Four samples are taken from each replicate and averaged. The graphical representation of the results obtained from proliferation is performed using the Prism 9 software (GraphPad Software, v.9.3.1., Boston, MA, USA). The percentages of cell proliferation are calculated for each replicate (previously sorted) as follows:$${Proliferation\;\left(\%\right)}_{i}=\frac{Cells\;number\;({Exp}_{i})}{Cells\;number\;({Control}_{i})}\times 100$$

The data are represented as a value of increase or decrease in proliferation as follows:$${Proliferation\;(\%)}_{j}={Proliferation\;(\%)}_{i}-100\%$$

The experimental design can be observed in Figure 4.

### 2.6. Apoptosis Assay

For apoptosis assays, the fluorescence Viability/Cytotoxicity Assay Kit for Animal Live & Dead Cells (Biotium, cat.no. 30002, Fremont, CA, USA) is used, which stains live cells green using Calcein AM and apoptotic cells red with EthD-III, following the manufacturer’s specifications. Briefly, the culture medium is removed from each of the samples tested. Add 2 µM Calcein AM and 4 µM EthD-III in sufficient volume to cover the monolayer. Allow to stand for 30 min at room temperature in the dark. Fluorescence is observed with a LEICA DFC340 FX microscope (Danaher, Washington, DC, USA) and DFC Twain camera software (DFC Twain, v.6.9.0.107, Leica Microsystems, Heerbrugg, Switzerland). The Calcein signal was visualized with the FITC filter set, and the EthD-III signal was visualized with the Texas Red filter set. The images were analyzed with FiJi software (ImageJ2, v.2.14.0, Madison, WI, USA).

The experimental design can be observed in Figure 5.

### 2.7. Statistical Analysis

SPSS Statistics software (IBM SPSS Statistics^©^ Software, v.29.0.0.0.0., New York, NY, USA) is used for statistical analysis of the different assays. For viability assays, the values of the 8 replicates are entered as a percentage of viability. In the case of proliferation assays, the number of live cells and the number of dead cells from the 12 samples obtained by frequency and controls are entered. The normality of the data is evaluated beforehand with a 95% confidence interval. If the data show a normal distribution, the Student statistical significance t-test is performed. If the samples have similar variance (Levene’s test, *p* > 0.05), the bilateral significance value is determined with the Student’s t-test for equality of variances. If the samples do not have similar variance (Levene’s test, *p* < 0.05), the bilateral significance value is read in the Student t-test for samples with non-similar variances. If the samples have a nonnormal distribution, the nonparametric Mann–Whitney U test is used.

## 3. Results

### 3.1. Viability

First, the 0–100 Hz frequency range is scanned at 20 Hz intervals during exposure times of 24, 48, and 72 h. Figure 6 shows the graphs representing the viability percentages obtained for each of the cell lines and the frequencies explored.

In the first frequency range studied, it is observed that the pheochromocytoma cell model (PC12) shows a bioactive window of decreased viability between 60 and 100 Hz, centered at 80 Hz (24 h: −28.52 ± 5.51%; 48 h: −34.75 ± 1.61%; 72 h: −39.26 ± 1.19%; *p* < 0.001) (Figure 6A). Furthermore, another bioactive window appears for the exposure time at 24 h and 48 h, centered at 40 Hz (24 h: −21.05 ± 3.27%, *p* = 0.035; 48 h: −27.34 ± 2.12%, *p* < 0.001). In the melanoma model (B16F10), the results of viability deviation with respect to the controls obtained are highly dependent on the exposure time used for a constant frequency (Figure 6B). In this case, exposure times of 72 h generally show decreases in viability, with the greatest decrease being obtained at 40 Hz (−18.54 ± 1.44%; *p* < 0.001). On the other hand, 24 h generally shows an increase in viability, with the greatest increase at 20 Hz (41.25 ± 3.44%; *p* < 0.001). The SBR3 cell line shows a decrease in viability at low frequencies, such as 20 Hz (24 h: −18.92 ± 4.19%; 48 h: −24.26 ± 0.91%; 72 h: −15.17 ± 2.12%; *p* < 0.001), 40 Hz (24 h: −5.62 ± 3.07%, *p* = 0.001; 48 h: −12.57 ± 3.19%, *p* < 0.001; 72 h: −12.72 ± 3.11%, *p* < 0.001), and 60 Hz (24 h: −6.70 ± 7.63%; 48 h: −15.67 ± 1.74%; 72 h: −18.81 ± 4.09%; *p* < 0.001). However, its viability percentages increase for 80 Hz (24 h: 12.19 ± 3.42%, *p* < 0.001; 72 h: 11.94 ± 7.08%, *p* = 0.005) and 100 Hz (24 h: 11.74 ± 1.74%, *p* < 0.001; 72 h: 5.86 ± 4.15%, *p* = 0.005) (Figure 6C). In this case, the behavior according to the exposure times used is similar according to the frequency used, contrary to the response found in the melanoma model. In the triple negative breast tumor cell model (MDA-MB−231), the exposure time of 72 h produces a generalized decrease in viability, being more pronounced at 20 Hz (−15.20 ± 1.43%; *p* < 0.001), 40 Hz (−14.11 ± 1.25%; *p* < 0.001), and 100 Hz (−14.49 ± 2.27%; *p* = 0.002) (Figure 6D). Shorter exposure times, such as 24 h, show a generalized increase in viability, being the highest for frequencies of 20 Hz (19.52 ± 2.29%; *p* < 0.001) and 40 Hz (20.97 ± 5.80%; *p* < 0.001). As in melanoma, the results for the same frequency are highly dependent on the exposure time used. Finally, the 3T3 cells show a valley of decreasing viability values centered at 40 Hz in which the three exposure times converge to similar viability values (24 h: −29.79 ± 4.31%; 48 h: −24.87 ± 4.18%; 72 h: −26.80 ± 2.67%; *p* < 0.001) and reach a maximum at the 60 Hz frequency (72 h: 4.11 ± 1.33%; *p* < 0.001), with the opposite behavior between frequencies (Figure 6E).

The 30 and 50 Hz frequencies for each of the cell lines are then incorporated (Figure 7). In the pheochromocytoma cell model (PC12), it shows a decrease in viability compared to controls for the three exposure times at 30 Hz (24 h: −28.53 ± 4.86%; 48 h: −23.93 ± 3.32%; 72 h: −27.78 ± 3.77%; *p* < 0.001) and 50 Hz (24 h: −27.69 ± 2.08%; 48 h: −23.69 ± 0.81%; 72 h: −30.88 ± 2.74%; *p* < 0.001) tracing two bioactive windows (Figure 7A). This also happens in the breast tumor cell models, SKBR3 and MDA-MB−231. In the case of the melanoma cells, 30 Hz (24 h: 13.52 ± 1.77%, 48 h: 29.87 ± 3.98%, 72 h: 25.45 ± 1.59%; *p* < 0.001) produce an increase in viability over the controls. When exposed at 50 Hz, the viability percentages converge for all three exposure times to a value above 20% (*p* < 0.001). In the murine fibroblast cell model (3T3), both frequencies produce a decrease in viability, without a defined trace of the bioactive window.

After incorporation of the 45 and 55 Hz frequencies (Figure 8), the pheochromocytoma cell model further defines the bioactive window of decreased viability centered at 50 Hz. In the melanoma (B16F10) and fibroblasts (3T3) cell models, the viability results obtained at 45 and 55 Hz, do not show a new bioactive window, with values similar to the nearby frequencies. In the case of the SKBR3 cells, the window is centered above 55 Hz (24 h: −17.01 ± 1.29%; 48 h: −37.75 ± 1.46%; 72 h: −38.31 ± 4.69%; *p* < 0.001). In the breast tumor model MDA-MB−231, it is centered at 45 Hz (24 h: −18.48 ± 3.70%; 48 h: −27.92 ± 5.19%; 72 h: −32.02 ± 7.92%). In the 3T3 cells, it is centered at 45 Hz, intermediate values to those obtained for the frequencies of 40 and 50 Hz are obtained in 24 and 72 h (24 h: −25.93 ± 3.19%; 72 h: −19.71 ± 1.58%; *p* < 0.001), in the case of 48 h (−32.83 ± 2.62%, *p* < 0.001), the minimum viability is reached for the frequencies studied.

The results show the dependence of the cellular response in viability on the frequency used, but also on the exposure time and the cell model used. It is shown how the time dependence is less evident at some selected frequencies in which the viability value converges independently of the exposure time used and some cell models.

### 3.2. Proliferation

Once the frequency window in viability was determined, it was decided to examine the proliferative capacity of the cells at frequencies of 45, 50, and 55 Hz, as these are the frequencies in which most of the cells obtain greater changes in the viability percentages.

The MDA-MB-231 cells show increases in proliferation with respect to the controls for the three frequencies studied, 45 Hz (59.10 ± 3.90%; *p* < 0.001), 50 Hz (41.74 ± 8.27%; *p* < 0.001), and 55 Hz (47.60 ± 3.82%; *p* < 0.001). This is similar for the melanoma cell model (B16F10) although in this case, the increases are not as large as in the previous cell model, 45 Hz (36.81 ± 3.40%; *p* < 0.001), 50 Hz (4.29 ± 2.36%; *p* = 0.046), and 55 Hz (4.42 ± 2.37%; *p* = 0.042). In the case of the cells of the pheochromocytoma cell model (PC12), the results are highly dependent on the frequency applied, 45 Hz (−24.19 ± 0.43%; *p* < 0.001), 50 Hz (−5.06 ± 0.46%; *p* = 0.06), and 55 Hz (3.37 ± 0.84%; *p* = 0.024), as is the case for the non-tumor model of fibroblasts, 45 Hz (12.53 ± 8.14; *p* = 0.007) and 50 Hz (−19.96 ± 2.81%; *p* = 0.007). The SKBR3 model shows a decrease in proliferation regardless of the frequency used, 45 Hz (−11.05 ± 1.79%; *p* = 0.006), 50 Hz (−23.87 ± 5.35%; *p* < 0.001), and 55 Hz (−28.26 ± 4.37%; *p* < 0.001).

In relation to dead cells, as seen in Figure 9B, at 45 Hz, the B16F10 cells (−36.52 ± 5.68%; *p* = 0.001) and the pheochromocytoma cells (−22.27 ± 8.13%; *p* = 0.006) reduce statistically significantly. In the case of the 3T3 cells, this percentage increases compared to controls (44.78 ± 18.28%; *p* = 0.005). At 50 Hz, the percentage of dead cells in melanoma (48.15 ± 15.27%; *p* = 0.033) and MDA-MB-231 (56.94 ± 10.69%; *p* < 0.001) increase. This percentage is statistically significantly reduced for the SKBR3 cells (−40.22 ± 4.38%; *p* < 0.001). At 55 Hz none of the results is statistically significant.

### 3.3. Apoptosis

Figure 10 shows the fluorescence results obtained. Neither in viability (Figure 10M), proliferation (Figure 10N), nor the number of dead cells (Figure 10O), the melanoma cell model shows statistically significant results for any of the frequencies evaluated. In the case of the MDA-MB-231 cells, it shows significant results of increased viability for the frequency of 45 Hz (*p* < 0.001), 50 Hz (*p* = 0.048), and 55 Hz (*p* = 0.016). It also shows significant results of increased cell proliferation at 45 Hz (*p* < 0.001). The number of dead cells is not affected (*p* = 0.06). In the case of the pheochromocytoma cells, it shows a single statistically significant value of increased viability at 45 Hz (*p* < 0.001), although this same frequency produces a decrease in cell proliferation (*p* = 0.004). It is the same frequency that shows a considerable increase in the number of dead cells in this same cell line (*p* = 0.029). 

## 4. Discussion

The main hypothesis of this work is that there is a specific frequency–frequency/intensity combination (a code) of a magnetic field that allows for the reduction of the viability and proliferation of a tumor cell model. Furthermore, as a secondary hypothesis to this first one, it is established that these codes are dependent on the cell type used and, for this reason, tumor cells from different tissues are used during the experimentation and will be compared with the results obtained in a previous publication using tumor cells from tumor (glioblastoma, neuroblastoma) and nontumor (astrocytes) nervous tissue [65].

The results show that cell viability and proliferation behavior depend on the frequency and type of cells applied. However, in addition, they are also dependent on the exposure time, which modifies the results in a way that depends on the duration of the applied exposure. The 40–60 Hz frequency range causes, in melanoma cells, an increase or decrease in viability depending on the exposure time applied, except for 50 Hz, where their behavior converges to the same percentage for the three times used. In the PC12, SKBR3 and MDA-MB-231 cells, the tendency of cell behavior is to decrease viability, although the frequencies of 40, 55, and 60 Hz modify this behavior in a time-dependent manner. The fibroblast cell model shows a decrease in viability in the same frequency range. Therefore, it is shown that the cell behavior of the same cell lineage is dependent on the frequency/time combination in what is known as the “biological window model”.

Numerous researchers have studied frequency as a possible cause of resonance effects in the cell that could determine the responses found [68,69,70,71,72,73,74,75,76,77,78]. However, the tendency of many authors is to use a specific frequency and not a wide range of frequencies in cell experimentation, which allows this parameter to be considered unimportant, and its role in the experimental design to be relegated to second place [64]. This also occurs because, for years, the intentionality parameter has been considered as the sole cause of cellular effects derived from exposure to magnetic fields, inherited from the translation of the study of ionizing radiation to non-ionizing radiation [59]. The cell was then understood as an energy detector, in which only sufficiently high amounts of energy could cause cellular effects. Nevertheless, there are many publications showing cellular alterations resulting from exposure to weak ELF-EMFs [69,79,80,81,82,83].

The frequencies of 50 Hz [24,25,28,33,41,42,43,44,45,47,84,85,86,87,88,89,90,91,92,93,94,95,96,97,98,99,100] and 60 Hz [28,48,101,102,103,104,105,106,107] are usually the frequencies of preference when designing experiments, because they are those frequencies used in the electrical distribution network throughout the world. In this study, the frequency of 50 Hz is of great importance because, in practically all cell lines used, the most important results of decreased or increased viability and/or proliferation were obtained, coinciding with the previous publication [65].

The existence of codes in biology is not new. A code can be defined as “a correspondence between objects in two independent worlds that is implemented by objects in a third world called adapters” [108]. The existence of a multitude of codes has been determined in biology [108]. The best known is the genetic code, but it is not unique; many other organic codes have been found to exist in living systems, such as the metabolic code [109], sequence codes [110,111,112], the histone code [113,114,115,116,117], the sugar code [118], or the tubulin code [119,120,121,122]. Our hypothesis holds that a specific combination of exposure parameters made up of specific values of frequency and time, which form a code, manages to activate the cell. This code would function as a key that opens a lock understood as cellular structures, making the cell response to the stimulus take the form of a window, altering its behavior in basic cellular processes (viability, proliferation) when this response is maximal (Figure 11). In other words, only certain ranges of parameters would cause the cell to respond. Knowing these specific combinations would allow researchers to manipulate this cell behavior as desired. An obvious example would be the therapeutic application of magnetic fields. This biological window behavior has been widely reported in other published studies [59,69,80,81,82,123,124,125,126,127].

The question that arises is: what is the most decisive parameter in the cellular response? This question does not seem to be easy to answer. With respect to the results obtained, it could be concluded that there is no single determining parameter that alone achieves the desired effects in all cell lines. Perhaps the solution is not to think of one parameter as the conductor of the orchestra, but rather a combination of these parameters that is specific to each cell type. We would understand this combination of parameters as a code that the cell receives and sets in motion the different mechanisms of action.

It is known that biological systems, in this case, cellular systems, are systems with non-linear responses. In addition, the cell membrane is a good electrical insulator (ε ≈ 6), which has led to the idea that the most important component in the interaction between magnetic fields and cells is the magnetic one [123,125,126,128,129]. It has been determined that, in the presence of a magnetic field, the cell undergoes alterations at the micro- and macromolecular level. The site of occurrence of the cellular effects in the first instance could be the cell membrane [123,125,126,128,129]. The cell, through its cell membrane, communicates intracellularly and extracellularly through ion currents that respond in most cases to the presence of an electric current. Therefore, it is determined that the cell has a double function: as a sensor, to detect the differences in the chemical gradient and the electrical currents generated; as an effector, to transmit the electrical and chemical signals to the rest of the cells [54,77,125]. It is not difficult to think that an external magnetic field could alter the cellular environment and cause modifications in its vital processes, such as viability and proliferation. It has been established that there are three main ways in which magnetic fields interact with living systems: energy, matter, and information transfer [59,81]. Cells of any multicellular organism require, in order to develop vital functions such as growth and division, constant communication between them, based on signaling molecules that, in excitable cells, combine with bioelectrical phenomena [77]. Communication phenomena are basically membrane phenomena that are produced by transduction complexes (specific receptor proteins, transducer proteins, enzymes) that amplify the weak signal that is promoted by the binding of signaling molecules to their specific receptors on the extracellular side of the cell membrane [125]. As a consequence, effector molecules are generated at the cytosolic level (second messenger) that induce various metabolic changes [54]. Many studies support the idea that ELF-EMFs could interact directly with proteins and transducer and receptor enzymes of the cell membrane and produce an amplification of the outer signal [130,131,132,133,134,135,136,137,138].

The interaction of magnetic fields and cell membrane proteins is mainly due to electronic polarization, reorientation of dipole groups, and changes in the concentrations of charged species in the vicinity of charges and dipoles [54]. Biological systems are complex and inhomogeneous, with ionic and dielectric properties that are difficult to predict [54,125]. Membranes do not maintain the same composition in different cells, which makes it difficult to determine the biological effects of a given magnetic field exposure, since they will depend strongly on the shape and composition of the surface and the presence/absence of charged or dipolar groups [54]. Three characteristics of the interaction process between an ELF-EMF and the cell membrane have been proposed to be of particular importance for amplification of the effects [54]: nonlinearity or non-equilibrium [125,126,139,140], cooperative processes [1], and resonance [54,141]. Resonance is directly related to frequency.

The bioactive window hypothesis states that interactions between living systems and the magnetic field involve information transfer, with discrete levels of amplitude and frequency that cause the effects observed experimentally, and can trigger biochemical processes, ion binding, and signal transduction [60,69,79,80,82,83,123,125,127,142]. The only way to justify the existence of these windows is to assume that biological systems are not linear [59,68,69,80,81,82,126,139]. Considering that there are windows of frequency, time, and intensity also leads the authors to determine that different electromagnetic fields applied to different tissues may cause different effects [123,124,125,143]. That is, the same EMF applied to different cell types may cause different effects, and the same cell type may or may not be affected by several EMFs, with similar or totally opposite effects.

Although there is no interaction model to explain the cellular effects found, the results allow us to think of possible therapeutic applications in which there is a “window of opportunity” or “therapeutic window”, as that combination of exposure parameters that could alter a pathological process through a therapeutic action [124]. In addition, it is stated that exposure to magnetic fields in an out-of-equilibrium system, in a pathological state, could be more effective than those applied to systems in equilibrium or healthy controls [124]. The signaling pathways and molecules involved in cell viability and proliferation processes are of great importance in the pathological development of tumor processes. Knowing the specific code that allows us to decrease the levels of one molecule or another, and stop cell proliferation, for example, in a tumor model, is of great clinical relevance. As has been observed in the results presented, cells from tumor models show different response to nontumor models. Tumor cells seem to be more sensitive to the magnetic field, and this is explained by the modifications they present in the signaling pathways of cell proliferation and viability [3,4,144]. Calcium pumps and channels are usually modified in tumor cells [1,145]. These cells overexpress specific calcium channels that activate pathways, such as PI3K/Akt, Ras/MAPK, and NFAT, that promote cell proliferation and also regulate the action of cyclins and cyclin kinases causing cells to proliferate uncontrollably [145,146,147]. It also plays a key role in cell migration and invasion, so that, in tumors such as breast, prostate, and colon tumors, channels, such as TRP (Transient Receptor Potential), are overexpressed, increasing calcium concentration, and thus activating actin cytoskeletons and favoring cell migration and invasion [87]. Furthermore, it also regulates angiogenic factors, such as VEGF (vascular endothelial growth factor), which promotes the formation of new blood vessels. In tumor cells, there is also an accumulation of calcium in mitochondria that blocks apoptosis by modifying the Bcl-2 proteins [145,148]. ROS also regulate PI3K/Akt and Ras/MAPK pathways, favoring cell proliferation and migration, as well as activating apoptosis evasion pathways through Bcl-2 and acting on factors, such as NF-kB and HIF-1α, that favor angiogenesis, invasion, and cell viability [149]. These differential aspects of tumor cells versus non-tumor cells make it possible to think that a magnetic field with a given combination of exposure parameters can be applied to act differently on pathological cells versus non-pathological cells. The magnetic field could alter the second messengers of signaling pathways responsible for cell proliferation and viability, for example, by increasing Bcl-2 expression to promote tumor cell apoptosis, modulating the behavior of calcium channels, or regulating mitochondrial activity and ROS concentrations [4,7,150]. However, the magnetic field itself could also act as a second messenger by altering cell signaling pathways, just as all other chemical molecules do. Also, the effects on non-tumor cells, such as fibroblasts in this case, are very important for tumor development. In this case, the fibroblast cell model obtained mostly decreased viability results. These results are particularly important. Since fibroblasts are responsible in the tumor microenvironment for promoting proliferation, migration, and invasion of cancerous tumor cells, they secrete growth factors and cytokines that promote angiogenesis and modify the extracellular matrix facilitating metastasis, so being able to target their response with specific bioactive codes is also of great importance for the role they play in relation to tumor cells [151]. On the other side, finding a bioactive code that would increase their viability and proliferation could be very useful in the therapeutic application of magnetic fields in pathologies related to cellular aging and collagen loss. Deciphering the coded response combinations for the reduction of tumor cell proliferation and viability would open the way for the therapeutic application of the magnetic field, alone or in combination with traditional therapies, such as pharmacological, offering a non-invasive and targeted therapeutic alternative according to the type of tumor and patient treated.

### Limitations

The main limitation of this work is that there is no physico-mathematical mechanism to explain the results of ELF-EMF exposure in cellular models, so the search for explanations of the cellular results must be based on hypotheses.

The range of values of the exposure parameters are infinite, so a selection of these had to be made. The values that can be adopted by exposure parameters, such as frequency and exposure time, are infinite. In this research a selection of these has been made on the basis of the results that were previously published in [64,65,66]. This does not mean that parameter values that have not been explored cannot obtain better results or results totally different from those presented in this work. The final objective of the experimentation was to show the implication of the frequency in the cellular response in viability and proliferation, and the experimental design has been sufficient for this; however, larger ranges of these parameters could be explored in the future, in order to obtain patterns of cellular behavior based on the values of the magnetic field exposure parameters.

Cellular assays are performed mainly on some immortalized cell lines that are models of different tumors, so they are not derived directly from patient tumors or animal models, i.e., they are not primary cells. Primary cells have the advantage of originating directly from the tissue or organ under study and behave similarly to the tissue of origin. However, they have a very limited capacity for division and, therefore, they would not be optimal for the study of viability and proliferation processes. Their maintenance in culture is complex. Although their characteristics allow for a more precise physiological study, they are not ideal for large-scale or repetitive experiments. It could be expected that the behavior between an immortalized cell line and a primary cell line in the face of the same magnetic field would vary from the results presented.

The in vitro results allow for an approach to the problem, and are adequate for the study of the hypotheses raised in this research. However, they do not allow us to analyze the hypotheses presented in cells inserted in their natural physiological environment, disregarding the homeostatic compensatory mechanisms that could modify some of the behaviors observed in the results.

The cellular processes studied (viability, proliferation) show global results of the alteration of the final product of this process; however, with the results obtained, it is only possible to hypothesize about the alteration of the secondary molecules involved in the process, and how the ELF-EMFs alter their normal processes. The results are presented in a global form, and studies that analyze the molecular level in depth would be necessary to give a more detailed explanation of the effects that ELF-EMFs cause in cells.

## 5. Conclusions

It has been demonstrated that there are certain frequencies that can reduce the proliferation and viability of different tumor cell models. The cellular effects found depend largely on the frequency used, but also on the exposure time and the cell line used. The presented results seem to confirm the hypothesis that there is a specific combination of cell type-dependent parameters that allow the alteration of cell viability and proliferation processes. The cell could be understood as a sensor capable of discriminating between different magnetic fields. This is of great utility for future therapeutic applications. 

## Figures and Tables

**Figure 1 biomolecules-15-00503-f001:**
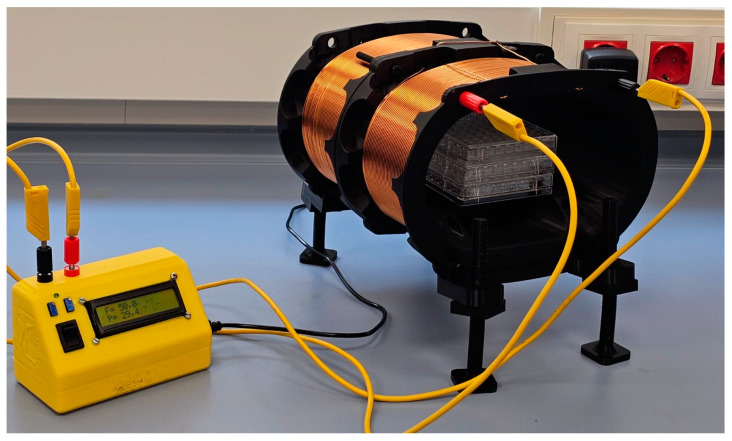
Magnetic field generation system, RILZ coils, specifically designed, manufactured and validated for cellular experiments.

**Figure 2 biomolecules-15-00503-f002:**
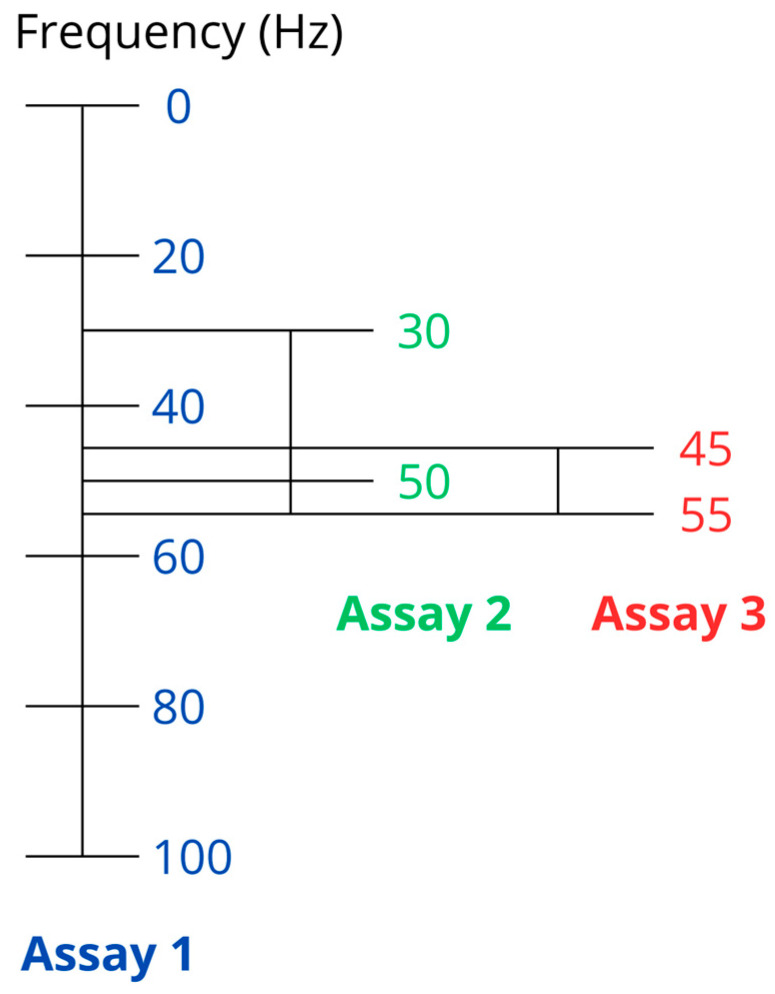
Graphical representation of the study of different frequency windows. Assay 1 evaluates the cell behavior in viability in a range of [20–100] Hz in 20 Hz ranges. In assay 2, the [20–60] Hz range is studied by reducing the step to 10 Hz. In assay 3, the range is determined at [40–60] Hz in 5 Hz steps.

**Figure 3 biomolecules-15-00503-f003:**
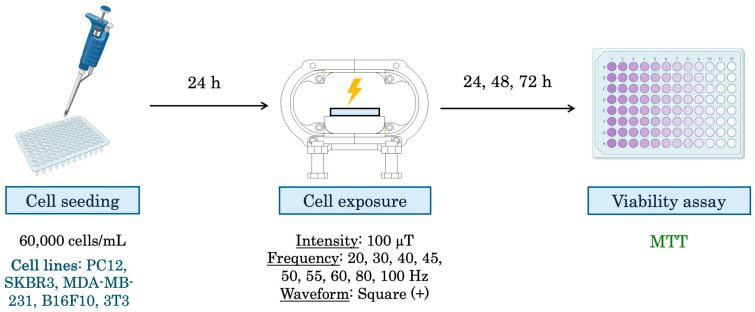
Experimental design for viability assays. First, cells are seeded, 24 h later cells are exposed using the RILZ system. After 24, 48, and 72 h of exposure, the cell viability test (MTT) is performed.

**Figure 4 biomolecules-15-00503-f004:**
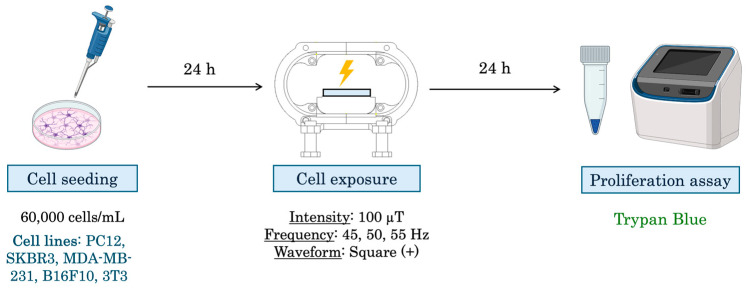
Experimental design for proliferation assays. First, cells are seeded, 24 h later cells are exposed using the RILZ system. After the selected exposure time has elapsed, the cell proliferation assay is performed using an automatic cell counter and the reagent trypan blue.

**Figure 5 biomolecules-15-00503-f005:**
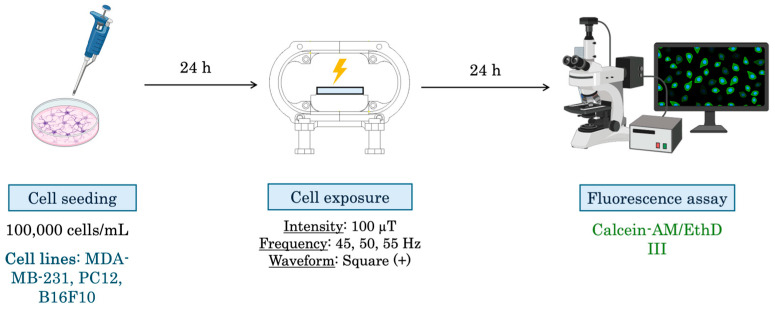
Experimental design for apoptosis assays. First, cells are seeded, 24 h later cells are exposed using the RILZ system. After the selected exposure time has elapsed, the cell apoptosis assay is performed using a fluorescence microscope.

**Figure 6 biomolecules-15-00503-f006:**
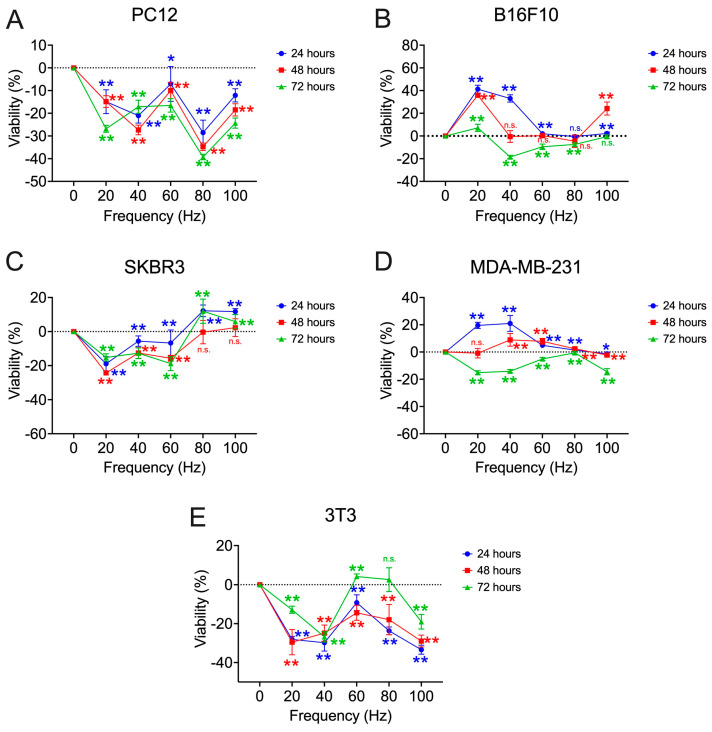
Percentage of viability deviation obtained in the search of the first frequency window [20–100] Hz of the different cell lines tested with respect to non-exposure controls: (**A**) rat pheochromocytoma (PC12); (**B**) murine melanoma (B16F10); (**C**) HER2+ human breast cancer (SKBR3); (**D**) triple negative human breast cancer (MDA-MB-231); (**E**) murine fibroblasts (3T3). All assays are performed at a fixed intensity of 100 µT at 24, 48, and 72 h of exposure. Statistical results from application of the Student t-test or the Mann–Whitney statistical U test with a 95%-CI according to normality of the data: (*) *p*-value < 0.05; (**) *p*-value < 0.001; (n.s.) non-significant.

**Figure 7 biomolecules-15-00503-f007:**
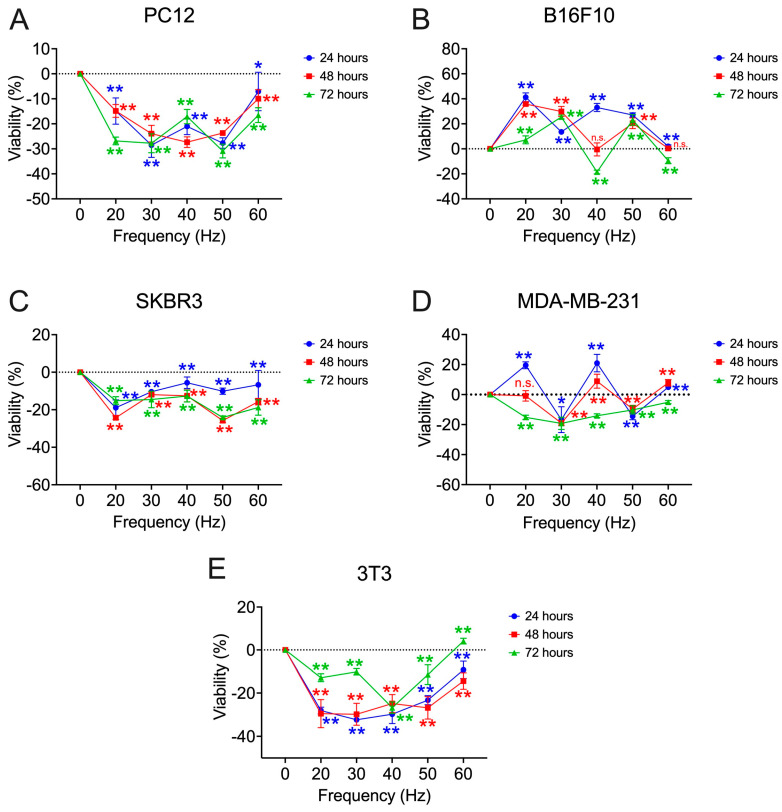
Percentage viability deviation obtained in the search of the second frequency window [20–60] Hz of the different cell lines tested with respect to non-exposure controls. The 30 and 50 Hz frequencies are incorporated: (**A**) rat pheochromocytoma (PC12); (**B**) murine melanoma (B16F10); (**C**) HER2+ human breast cancer (SKBR3); (**D**) triple negative human breast cancer (MDA-MB-231); (**E**) murine fibroblasts (3T3). All assays are performed at a fixed intensity of 100 µT at 24, 48, and 72 h of exposure. Statistical results from application of the Student t-test or the Mann–Whitney statistical U test with a 95%-CI according to normality of the data: (*) *p*-value < 0.05; (**) *p*-value < 0.001; (n.s.) non-significant.

**Figure 8 biomolecules-15-00503-f008:**
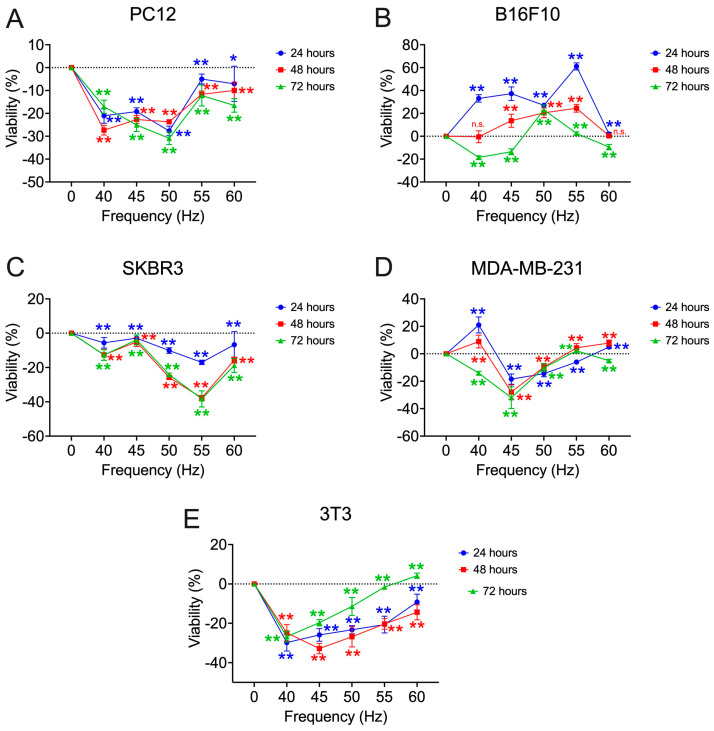
Percentage of viability obtained in the search of the third frequency window [40–60] Hz of the different cell lines tested with respect to non-exposure controls. The 45 and 55 Hz frequencies are incorporated: (**A**) rat pheochromocytoma (PC12); (**B**) murine melanoma (B16F10); (**C**) HER2+ human breast cancer (SKBR3); (**D**) triple negative human breast cancer (MDA-MB-231); (**E**) murine fibroblasts (3T3). All assays are performed at a fixed intensity of 100 µT at 24, 48, and 72 h of exposure. Statistical results from application of the Student t-test or the Mann–Whitney statistical U test with a 95%-CI according to normality of the data: (*) *p*-value < 0.05; (**) *p*-value < 0.001; (n.s.) non-significant.

**Figure 9 biomolecules-15-00503-f009:**
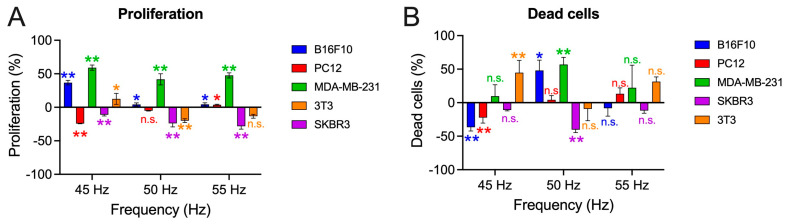
(**A**) Percentage of proliferation of the different cell lines tested with respect to the non-exposure controls. (**B**) Percentage of dead cells with respect to non-exposure controls. All tests are performed at frequencies of 45, 50, and 55 Hz with a fixed intensity of 100 µT in 24 h of exposure. Statistical results from application of the Student t-test or the Mann–Whitney U test with a 95%-CI according to normality of the data: (*) *p*-value < 0.05; (**) *p*-value < 0.001; (n.s.) non-significant.

**Figure 10 biomolecules-15-00503-f010:**
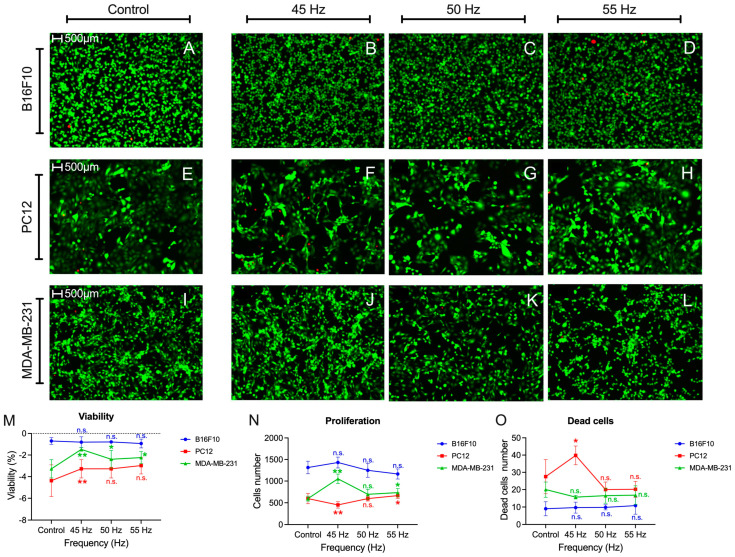
Apoptosis assay results at different frequencies of 45 Hz (**B**,**F**,**J**), 50 Hz (**C**,**G**,**K**) and 55 Hz (**D**,**H**,**L**) for B16F10 (**A**–**D**), PC12 (**E**–**H**) and MDA-MB-231 (**I**–**L**) tumor cell line models. The results of viability (**M**), proliferation (**N**) and dead cells (**O**). Statistical results from application of the Student t-test or the Mann–Whitney statistical U test with a 95%-CI according to normality of the data: (*) *p*-value < 0.05; (**) *p*-value < 0.001; (n.s.) non-significant.

**Figure 11 biomolecules-15-00503-f011:**
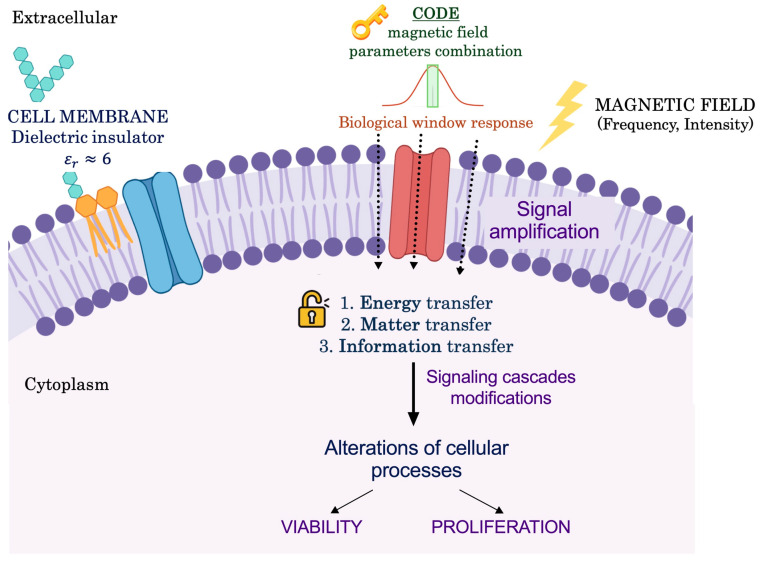
Hypothesis of interaction of the magnetic field and a cell. The cell receives the magnetic field stimulus as an encoding of exposure parameters that is in a bioactive range for a given cellular process. The first site of interaction is the cell membrane, where signal amplification occurs, resulting in a transfer of energy, matter, and information. This alters the cellular process through interaction with specific molecules of the cell signaling cascades. This leads to alteration of the basic cellular processes of viability and proliferation.

**Table 1 biomolecules-15-00503-t001:** List of exposure parameters (intensity, frequency, and exposure time) and biological parameters (cell line, biomarkers) used in each of the experiments presented.

Assay No.	Cell Line	Intensity [µT]	Frequency [Hz]	Exposure Time [Hours]	Biological Assay
1	PC12, SKBR3, MDA-MB-231, B16F10, 3T3	100	20, 40, 60, 80, 100	24, 48, 72	Metabolic activity (MTT)
2	PC12, SKBR3, MDA-MB-231, B16F10, 3T3	100	30, 50	24, 48, 72	Metabolic activity (MTT)
3	PC12, SKBR3, MDA-MB-231, B16F10, 3T3	100	45, 55	24, 48, 72	Metabolic activity (MTT)
4	PC12, SKBR3, MDA-MB-231, B16F10, 3T3	100	45, 50, 55	24	Proliferation and number of dead cells (Trypan Blue)
5	PC12, SKBR3, MDA-MB-231, B16F10, 3T3	100	50	24	Viability/Apoptosis (Calcein/EthD)

## Data Availability

The raw data supporting the conclusions of this article will be made available by the authors on request.

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
