# Peer review of "The Frequency of a Magnetic Field Reduces the Viability and Proliferation of Numerous Tumor Cell Lines"

_biomolecules, 2025, doi:10.3390/biom15040503_

Round 1
Reviewer 1 Report
Comments and Suggestions for Authors
This research articles describes " The frequency of a magnetic field reduces the viability and 2 proliferation of numerous tumor cell lines "
This research article is applying a novel physical technique as a novel treatment method for cancer.
The authors tried to investigated the effect of frequency and exposure time in the behavior of normal and cancerous cells.
The study is unique and novel and it will be very helpful if the authors should explain the following comments………
- What is full name of ELF-EMF ? the author should write the full name of ELF-EMF for the first time where this abbreviation appears
- The should report the advantages of using a magnetic field to reduce the viability and proliferation of different tumor cell models compared to the conventional or the approved methods
- What are the effect of the applied magnetic field in the normal cells?
the English language needs slight improvement
Reviewer 2 Report
Comments and Suggestions for Authors
The authors show their study on different cell lines submitted to magnetic exposure and the effect on the viability, proliferation in cancer and non cancer cells. This is a very interesting topic that could be very useful thinking of cancer treatment. I address some comments bellow to make clear some of the questions that were raised during the reading.
- Line 21: I think the authors should mention since the beginning in the abstract the type of cell line. Mention which ones are human and which ones are from mouse or rat.
- Line 36: Write the meaning of ELF-EMF.
- Regarding the cell culture, why it was chosen different type of cell lines from different tissues and different species? I found it on the discussion section but it should be mentioned also before showing the results.
- Line 150: I suggest writing that it is exposure time, to be clear regarding the experimental steps. The same applies to the table.
- I also suggest making a scheme of table 1. It is the design of your experiment, but it can take time to understand. If you make a scheme showing 1st experiment variables, then that in sequence (arrow) new variables were studied in the second experiment, showing that the intention was to narrow the frequency window. This could be easily done in the table.
- For the figure 4 it was used the range of 20-100 Hz, with the variables of 20, 40, 60, 80, 100. In the figure 5, if it is following the table 1, why there are more frequencies included? I think in the legend of figure 5 and 6 it should be written as in line 348 “In the last range explored, the XX and XX Hz frequencies are incorporated”. It should also be clear in table 1.
Reviewer 3 Report
Comments and Suggestions for Authors
Please find attached the review report

Round 2
Reviewer 3 Report
Comments and Suggestions for Authors
Accept in current form